# OpenReview forum: "Understanding Square Loss in Training Overparametrized Neural Network Classifiers"
_ICLR.cc/2022/Conference — ICLR 2022 Submitted_

### Official Review · Reviewer_MNxz · 2021-11-01

**Correctness:** 3
**Technical Novelty And Significance:** 3
**Empirical Novelty And Significance:** 2
**Recommendation:** 5
**Confidence:** 2

**Main Review:**

**Pros**
+ The paper improves on previous results on the convergence of neural networks, by deriving a faster convergence rate (however under stronger assumptions).

+ In contrast to prior work, which assumes data to be linearly separable, the paper studies the more general and much more realistic setting of separable data (and also of unseparable data).


**Cons** (see also detailed comments below)
- The presentation of the theoretical results is superficial and crucial information can only be found in the appendix (see detailed comments below)

- The content and setting of the experiments differs from the theoretical contributions and thus they do not corroborate them.

- There are some claims in the paper, which need more backing.


**Detailed comments:**
- Presentation of the results:

  - Crucial information is missing from the main text and can only be found in the supplementary material, most importantly, the assumptions under which the theorems hold. Furthermore, these assumptions are only stated, but not discussed. In consequence, it remains unclear, whether these assumptions are, for example, fulfilled in practice, might be loosened, or and where they come from.

  - The theoretical results are not properly stated as they do not specify what the respective variables mean. This becomes a real issue, when the variables have not been defined in the text so far (or at any later point).
For example, Theorem 3.1 is a asymptotic equality for some function $\hat f$, which is not specified in the main text. In fact, $\hat f$ is only defined in the passing, somewhere in the supplementary material as a solution of some minimization problem stated one page earlier. However, then one needs to look up the function space on which the minimization occurs, which in turn is specified two pages earlier. But this is only helpful, if the reader already knows, what the "reproducing hilbert space generated by the neural tangent kernel on $\mathbb S^{d-1}$" is.  This is clearly suboptimal.
Similarly, Theorem 3.2 contains a summand $\delta$, which is not defined anywhere, and appears to can be chosen $=0$ anyway.
Theorem 3.6 contains  the imprecise formulation "no matter how complicated $\Omega_1 \cup \Omega_2$ are". Does this mean for any $\Omega_1 \cup \Omega_2 \subset \Omega$?. It would also improve the readability, if the definitions of $\Omega_1$ and $\Omega_2$ are restated in the theorem or in its proximity.

- Experiments:
  - The experiments are somewhat disconnected from the theoretical part. In the experiments, a comparison between the squared loss and the cross entropy loss is made, similar to the empirical study [1]. However, the theory exclusively covers properties of the square loss, and thus does not make any statement or prediction whether training with either of these loss functions is preferable. From my perspective, a comprehensive on the extent to which the theoretical predictions hold would have been more valuable.

  - Similarly, the experiments are not designed according to the presented theory. The theory mainly treats  1-hidden-layer networks, whereas experiments use either 2-hidden layer experiments (5.1) or ResNets (5.2). Furthermore, the 500 hidden neurons in (5.1) do not seem to be wide enough, such that the assumptions of the theorem are fulfilled.

- Not sufficiently supported claims:
  - Remark following Theorem 3.4
"Under {0,1} coding, the estimator would be $f_{W,A}$ itself. The ${L_\infty}$ consistency doesn’t hold for cross-entropy trained neural networks, due to the form of the optimal solution $\log(\frac\eta{1-\eta})$. With limited capacity, the network’s confidence prediction is bounded away from 0 and 1."
This needs to be discussed in more detail, as Theorem 3.4 is of asymptotic nature (infinite width), but the presented argument only rules out networks with limited capacity.

  - Regarding the simplex label coding
  "The one-hot coding forces the features for each class to be orthogonal in the unit sphere, which is problematic for large K since the representations are all cramped into the corner (positive direction in each dimension) taking only $2^{-K}$ of the entire sphere.
In comparison, maximally separated K points on the unit sphere is the better choice, which coincides
with the vertices of a (K-1)-simplex in K-1 dimensions."
From my understanding, this not an issue. As has already been shown in the neural collapse literature, the one-hot encoding leads (at optimum) to a simplex coding in the penultimate layer. See for example [2].


- Misc.
  - Proposition 4.2 has already been proven in [3] and in a more general case in [4]. Furthermore, here the proof appears to only cover the special case of 2 instances per class.

  - The train and test data in Figure 1 are very difficult to distinguish.

References:
[1] Like Hui and Mikhail Belkin, Evaluation of neural architectures trained with square loss vs cross-
entropy in classification task, ICLR 2021
[2] DG Mixon, H Parshall, J Pi, Neural collapse with unconstrained features. arXiv:2011.11619
[3] C. Fang, H. He, Q. Long, and W. Su, Exploring deep neural networks via layer-peeled model: Minority
collapse in imbalanced training, Proceedings of the National Academy of Sciences, 2021
[4] Florian Graf, Christoph Hofer, Marc Niethammer, and Roland Kwitt, Dissecting supervised contrastive
learning, ICML 2021

**Summary Of The Paper:**

The paper studies wide neural networks trained with the square loss in the neural tangent kernel regime from a theoretical perspective. In particular, it provides generalization error bounds, robustness and calibration results and makes a connection to supervised contrastive learning. Furthermore, an empirical comparison between cross entropy and square loss is provided.

**Summary Of The Review:**

I am not very familiar with the related work and thus cannot really evaluate the papers significance and potential impact. That being said, the paper is difficult to navigate, as crucial information lies in the appendix and the experiments do, from my perspective, not really corroborate the theoretical contribution. Therefore I cautiously give a score of 5.

---

> ### Author Response · Authors · 2021-11-15
> **Response to Reviewer MNxz**
>
> We thank you for the helpful comments and suggestions. Below we address the detailed comments in the review
>
> ###  Presentation of results:
>
> - We thank you for the suggestions and comments. Due to page limitations, we chose to put some typical technical details (those frequently used in other papers) in the appendix so that the reader can have a smoother reading experience. In the updated submission, we have clarified the notation ($\hat{f}, \delta, \Omega_1, \Omega_2$, etc.), added a brief overview of reproducing kernel Hilbert space, revised the statements of the theorems, and added detailed discussion about our technical assumptions (in Appendix D).
>
> - The term $\delta$ only depends on the width of the neural network. A smaller $\delta$ requires a wider neural network. If $\delta=0$, then the number of nodes in the hidden layer is infinity. In the separable with positive margin case, our Theorem 3.2 and 3.6 don't put any assumptions on $\Omega_1$ and $\Omega_2$. The support of the two classes ($\Omega_1$ and $\Omega_2$) can be arbitrary as long as they are separated by a positive margin, which is much weaker than assuming linear separable.
>
> ### Experiments
>
> - We agree that our numerical simulation does not exactly match the assumption of our theories. As much as we would like to develop convergence rates under more practical settings (proper size with all the training tricks), the technical difficulties are too great. Although the setting of NTK is restrictive, we do believe they provide valuable insights beyond its scope and the properties revealed will carry over to actual neural networks.
>
> - For comparison between CE and SL, and CE not being consistent as estimating $\eta$, please see the Overall Responses (Part 3) for details.
>
> ## Not sufficient claims
> - The discussion about $L_\infty$ consistency of square loss and cross-entropy can be found in the Overall Response (Part 1). We have also made corresponding modifications and additions to the submission to make it more clear.
>
> - We thank the suggested reference related to the simplex coding. We have deleted the simplex coding discussion and made direct citations to existing works.
>
> ## Misc:
>
> Thank you for the suggestions. We have made the changes accordingly. Specifically, Figure 1 is plotted with different colors.

---

> > ### Comment · Reviewer_MNxz · 2021-11-18
> > **Response to Authors**
> >
> > Thank you for your response.
> >
> > The notation in the updated manuscript is now a bit more clear. Also the $L_\infty$ issue was adressed.
> >
> > However, the paper is still difficult to read, as crucial information can only be found in the supp mat. Out of the assumptions, only assumption D.1 is informally stated/discussed in the main text.
> > Furthermore, when reading the theorems, one needs to go on a treasure hunt for finding the definitions of the involved variables.
> >
> > I still think that the experiments are disconnected from the theory.
> >
> > I will keep my evaluation.

---

### Official Review · Reviewer_RzXw · 2021-11-01

**Correctness:** 3
**Technical Novelty And Significance:** 3
**Empirical Novelty And Significance:** 3
**Recommendation:** 5
**Confidence:** 4

**Main Review:**

Pros:
- The non-parametric fast rate under the Tsybakov noise condition in the NTK Regime.
- Results of model calibration are impressive.
- some theory and experiment have gap

Weakness:
- No comparison between L2 and cross-entropy loss
- lack of reference

Detailed comments:

**In terms of theory** The paper investigated standard setting of low noise condition (theorem 3.1) and achieves the exponential convergence rate for the separable case. The most interesting part is Theorem 3.4 (Calibration error). However, Theorem 3.4 doesn't have the tsybakov's noise condition kappa in the final convergence bound. Is it standard in this setting? Is it tight? It's slower than the convergence rate of the Excess risk?

**Simplex label coding** I guess most of the prove have been done by the neural collapse literature [7,8] , and it's also true for the CE loss [9].

The paper gives positive results for L2 losses,however, the paper didn't show any results that the cross-entropy loss is not good. The convergence of CE loss can be slow in the terminal phase of deep learning training [6] (O(1/log t)), but if you train longer, maybe all the properties you proved can appear. I hope the author can provide that why CE loss is not good enough.

**Adversarial robustness** The gain of SL interms of the adversarial loss is actually not significant, I would to suggest the author to try more norms(l2 norm robustness as example) can be tried in this setting. In the experiment I can't see what's the norm the author is using for PGD.(The number here looks like l_inf norm, but in theory theorem 3.3. it's l2 norm. I see a mismatch here.) For l2 robustness, you can further try randomized smoothing [5].


**Missing Reference** Optimal rate for NTK for regression [1,2] and outof NTK regime (but no optimization)[3,4], both of the papers using l2 loss.

[1] Nitanda A, Suzuki T. Optimal rates for averaged stochastic gradient descent under neural tangent kernel regime[J]. arXiv preprint arXiv:2006.12297, 2020.

[2] Hu T, Wang W, Lin C, et al. Regularization Matters: A Nonparametric Perspective on Overparametrized Neural Network[C]//International Conference on Artificial Intelligence and Statistics. PMLR, 2021: 829-837.

[3] Schmidt-Hieber J. Nonparametric regression using deep neural networks with ReLU activation function[J]. The Annals of Statistics, 2020, 48(4): 1875-1897.

[4] Farrell M H, Liang T, Misra S. Deep neural networks for estimation and inference[J]. Econometrica, 2021, 89(1): 181-213.

[5] Cohen J, Rosenfeld E, Kolter Z. Certified adversarial robustness via randomized smoothing International Conference on Machine Learning. PMLR, 2019: 1310-1320.

[6] Lyu K, Li J. Gradient descent maximizes the margin of homogeneous neural networks[J]. arXiv preprint arXiv:1906.05890, 2019.

[7] Fang C, He H, Long Q, et al. Layer-peeled model: Toward understanding well-trained deep neural networks[J]. arXiv preprint arXiv:2101.12699, 2021.

[8] Zhu Z, Ding T, Zhou J, et al. A Geometric Analysis of Neural Collapse with Unconstrained Features[J]. arXiv preprint arXiv:2105.02375, 2021.

[9] Ji W, Lu Y, Zhang Y, et al. An Unconstrained Layer-Peeled Perspective on Neural Collapse[J]. arXiv preprint arXiv:2110.02796, 2021.



**Summary Of The Paper:**

This paper considers the non-parametric convergence rate of neural network (NTK regime) under tsybakov's noise condition (low noise with large margin) and discussed the potential application of the theoretical result on the robustness of neural network and calibration results.

**Summary Of The Review:**

my main two concern:
- missing discussion of the relationship between line of research of using NN as a non-parametric estimator.
- I don't see any discussion of the specialty of the l2 loss. The paper don't answer the question of how l2 loss and CE loss is different.

---

> ### Author Response · Authors · 2021-11-15
> **Response to Reviewer RzXw**
>
> We thank you for the helpful comments and suggestions.
>
> For comparison between L2 and cross-entropy loss, please see the Overall Response (Part 1) for the discussion. We have also made corresponding modifications and additions to the submission. Below we address the detailed comments in the review.
>
> **In terms of theory**: This is because the calibration error is more subtle than the misclassification error, and puts a stronger requirement on the estimator. Consider the following example. Let $\eta(x)=P(Y=1|X=x)$. Suppose $\eta(x_1)=0.8$. Then the misclassification error is the same if two estimators are $\tilde f_1(x_1)=0.6, \tilde f_2(x_1)=0.79$. However, the calibration errors are very different. We need to accurately estimate all probabilities in order to make the calibration error small, and it does not depend on whether the probability is close to 0.5 or not. Tsybakov's noise condition only measures the probability of $\eta(x)$ around 0.5, which is crucial to the misclassification error, but not the calibration error. Therefore, it is slower than the convergence rate of the excess risk and it does not depend on the component in Tsybakov's noise condition.
>
> **Simplex label coding:** Thank you for bringing the reference. Indeed, our argument about simplex coding has been discussed in the literature. We have rewritten Section 4 and added the existing work.
>
> **Adversarial robustness**: The experiments are all in $l_\infty$ norm. Theorem 3.3 is on L_2 norm but since L_2 norm lower bounds L_infty norm, similar lower bound also holds for $l_\infty$ attacks. The improvement from 1% to 5% may not be that significant but the same margin can be carried over to actual adversarial training. We refer to the table in Overall Response (part 3) for details.
>
> **Missing references:** We appreciate your suggestions and include a paragraph in Appendix A to discuss the literature within and out of the NTK regime. Specifically, we state
>
> "*There are extensive works studying the generalization error bounds under NTK regime. For regression, [Nitanda & Suzuki 2020, Hu et al. 2021] show the optimal convergence rates when using overparametrized one-hidden-layer neural networks,  where the square loss is used.   [Arora et al.2019] provides generalization error bounds and provable learning scenarios for noiseless data.  In the NTK regime, the neural network as a regressor is linked with the nonparametric regression viaNTK. There are also other works studying the generalization performance of the neural network as a nonparametric regressor, out of the NTK regime; see [Schmidt-Hieber 2020, Farrell et al. 2021].*"

---

> > ### Comment · Reviewer_RzXw · 2021-11-25
> > **Thanks for your response**
> >
> > I've read the response, the discussion between l2 and CE loss is kind of interesting, hopes this can be included in the paper.
> >
> > **According to Adversarial robustness:** Yes, L_2 norm lower bounds L_infty norm, but there is a large dimension related constant which is not the realistic setting in my mind. I think the L_2 norm adversarial robustness is also an interesting topic (used in many papers) so that the author can simply change the norm projection simply in the attack code.
> >
> > I think this paper is an interesting paper but still not so much theoretical contribution based on the previous works. I'll keep my score at 6.
> >
> > P.s. I just find out related using non-parametric estimation under tsybakov noise  using NN, maybe good reference for authors
> >
> > [1] Bos T, Schmidt-Hieber J. Convergence rates of deep ReLU networks for multiclass classification[J]. arXiv preprint arXiv:2108.00969, 2021.
> >
> > [2] Kim Y, Ohn I, Kim D. Fast convergence rates of deep neural networks for classification[J]. arXiv preprint arXiv:1812.03599, 2018.
> >
> > [3] Kohler M, Langer S. Statistical theory for image classification using deep convolutional neural networks with cross-entropy loss[J]. arXiv preprint arXiv:2011.13602, 2020.

---

### Official Review · Reviewer_Eo56 · 2021-11-03

**Correctness:** 4
**Technical Novelty And Significance:** 3
**Empirical Novelty And Significance:** 3
**Recommendation:** 6
**Confidence:** 4

**Details Of Ethics Concerns:**

NAN

**Main Review:**

Strengths:
1.	The theoretical analysis of this paper is rich, including generalization and robustness theory.
2.	It fully verifies the theoretical advantages of square loss in several perspectives and provides the corresponding experimental verification for theoretical results in the neural tangent kernel (NTK) regime.

Weaknesses:
1.	When introducing the theoretical results, we should make a detailed comparison with the existing cross-entropy loss results. The current writing method cannot reflect the advantages of square loss.
2.	The synthetic experiment in a non-separable case seems to be a problem. Considering the nonlinear expression ability of neural networks, how to explain that the data distribution illustrated in Figure 1 is inseparable from the network model?
3.	This paper presents that the loss functions like hinge loss don’t provide reliable information on the prediction confidence. In this regard, there is a lack of references to some relevant literature. [Gao, 2013] has given a detailed analysis of the advantages and disadvantages between the entire margin distribution and the minimum margin. Based on this, [Lyu, 2018] designed a square-type margin distribution loss to improve the generalization ability of DNN.

[Gao, 2013] W. Gao and Z.-H. Zhou. On the doubt about margin explanation of boosting. Artificial Intelligence 203:1-18 2013.

[Lyu, 2018] Shen-Huan Lyu, Lu Wang, and Zhi-Hua Zhou. Improving Generalization of Neural Networks by Leveraging Margin Distribution. http://arxiv.org/abs/1812.10761



**Summary Of The Paper:**

This paper provides a theoretical guarantee for square loss in the neural tangent kernel (NTK) regime. Whether classes are separable or not, the convergence rate is proved to be improved.

**Summary Of The Review:**

In a word, I agree with the argument put forward in this paper and think its theoretical conclusion is correct. If the author can provide the theoretical comparison between SL and CL and supplement some missing citations of related work, I will more recognize the integrity of the paper.

---

> ### Author Response · Authors · 2021-11-15
> **Response to Reviewer Eo56**
>
> We thank you for the helpful comments and suggestions. Below we address the mentioned weaknesses in the review.
>
> 1. Please see the Overall Response (Part 1) for the discussion on Cross-Entropy vs Square Loss. Following your suggestion, we included a brief discussion after Theorem 3.4 (calibration error) to state the reason why cross-entropy loss does not lead to an accurate estimator of the conditional probability.
>
> 2. We would like to point out that Figure 1 is for the separable case because there exists a curve that can split the two supports. Here the separable or not depends on whether $P(Y=1|X)$ can take value other than {0,1} or in other words, whether the labels contain noises.
>
> 3. Thank you for the suggested references. They are interesting but for the predicted confidence, our definition of confidence is different from that defined in [Gao 2013] and [Lyu 2018]. The margin in our work refers to the distance between the decision boundary ($\{x: \hat{f}(x)=0\}$) and the class supports $ \Omega_1, \Omega_2$ while the margin in [Gao 2013] and [Lyu 2018] measures the gap between the largest and the second-largest predicted confidence. The predicted conditional probability $\hat{\eta}$ in Proposition 4.1 provides a **consistent** estimation of the ground truth while the "confidence" defined in [Gao 2013] and [Lyu 2018] does not.

---

> > ### Comment · Reviewer_Eo56 · 2021-11-22
> > **Response to Authors**
> >
> > Thank you for your response.
> >
> > However, the author's answer on the comparison between CE and SL did not improve the importance of this work, so I will keep my evaluation.

---

### Official Review · Reviewer_2eZX · 2021-11-03

**Correctness:** 3
**Technical Novelty And Significance:** 3
**Empirical Novelty And Significance:** 3
**Recommendation:** 6
**Confidence:** 3

**Main Review:**

Pros:
+ New convergence results and insights of squared loss for classification.
+ The results hold for a non-separable case using the Tsybakov noise condition.
+ Additional insights on the robustness and model calibration.

Cons:
- The analysis under the NTK regime is somewhat standard and may not be practical. For example, when mu=0, GD operates under the kernel regime, which do not explain some phenomenon observed in practice such implicit bias.
- There are a number of restrictive assumptions, for example, C.1, C.2 and C.3. In linear regression (Du et al., 2019), C.1 is sufficient, and C.2 seems to be sufficient for logistic regression (Ji et al. 2020).
- The experimental results on real data reinforce but perhaps repeat previous work.


Questions:
1. When is the definition of separability (when \eta only takes either 0 or 1 in two separate sets) more general than the linear separable case?
2. How does this result compare to other results of implicit biases of gradient descent for separable and non-separable data? Similarly, how does the convergence rate compare to that of Ji et al., 2020, which uses logistic loss and seems to assume less restrictive assumptions? A discussion would be helpful for readers.
3. It looks like the regularization term is a crucial part in the analysis important for this analysis. Does the analysis carry over if \mu is exactly zero?
4. Paragraph under Theorem 3.1: where is the (d-1)\kappa that you’re mentioning?
5. Assumption C.1: why does delta_n depend on n? When n=1, Eq. (A.3) implies that it is positive, right? Also, \lambda_0 scales with poly(1/n), when n -> infty, is \lambda_0 = 0?
6. Can you say anything about the robustness of squared loss to outliers? Empirical evidence is sufficient. Assumption C.3 seems to be related to that.

Minor:

1. Equation (2.1): the first expectation is taken over both (X, Y) ~ P? What does expectation over f(X) >= 1 mean? It can be re-written over X in a more precise way.
2. Please consider using \mathbf{a} instead of capital A for the second layer weight because the latter can be easily confused with a matrix. The objective in (2.2) gives the impression that A is optimized, but it is not. The analysis would be very different with a joint optimization over both W and A.
3. The color bar in Figure 1: how probability P[y=1] can be negative? I don’t see any color difference in each of the plots.

**Summary Of The Paper:**

This paper provides a theoretical analysis of the squared loss for classification where cross-entropy loss is often the standard choice in both theory and practice. The authors show under some assumption of the data and the NTK regime, GD on squared loss enjoys fast convergence, strong robustness and good calibration. They also provide empirical results.


**Summary Of The Review:**

See above

---

> ### Author Response · Authors · 2021-11-15
> **Response to Reviewer 2eZX (Part 2)**
>
>
>
> ## We address your questions as follows.
>
> Q1: Our separability assumption is more general than linear separation, just like nonparametric regression is more general than linear regression. Take Figure 1 for example, the classes are separable and not linearly separable. Note that the definition of the separability is only on the input space, not the space derived by some particular (predefined) feature map. For example, if $d=2$, then two supports are separable as long as there exists a **curve** that can separate the two supports, no matter how complicated the curve is.
>
> Q2: In [Ji & Telgarsky 2020], two supports are also separable with positive margin. The generalization error bound in [Ji & Telgarsky 2020] is in the order of $n^{-1/2},$ which is **much slower** than ours: we derived a generalization error bound with **exponential** rate. Even for the non-separable case, our generalization error bound can be faster than $n^{-1/2}$, when the Tsybakov noise component $ \kappa$ is large. Following your suggestion, we add a discussion after Theorems 3.1 and 3.6.
>
> Q3: Yes, the regularization term is crucial, when the supports are non-separable (and the noise exists). As suggested by [Hu et al. 2021], if noise exists, then regularization matters. [Hu et al. 2021] also apply a regularization term to address noisy labels, where the labels are randomly flipped. However, if the supports are separable, one can directly take $\mu =0$, which is also reflected in Theorems 3.2 and 3.3,  where we only require $\mu=o(1)$.
>
> Q4: Thank you for pointing it out. Note that in Theorem 3.1, the rate is $n$ to the power of $-\frac{d(\kappa+1)}{(2d-1)(\kappa + 2)}$, which is $n$ to the power of $-\frac{d(\kappa+1)}{(d-1)\kappa + d\kappa +4d- 2}$ while the optimal rate is $n$ to the power of ${-\frac{d(\kappa+1)}{d\kappa+4d-2}}$. Therefore, there is a $(d-1)\kappa$ difference in the denominator.
>
> Q5: $\delta_n$ depends on $n$ because we would like to have $\delta_n\rightarrow$0  such that the $O_p$ in Theorems 3.1 and 3.4 hold. You are right that it is strictly positive when $n=1$. You are also correct that as $n\rightarrow \infty, \lambda_0\rightarrow 0$. However, since $\lambda_{\min}(\mathbf{H}^\infty)>0$ with probability one, we can take $\lambda_0$ strictly positive (possibly depending on $n$). In fact, there is a trade-off between $\delta_n$ and $\lambda_0$: if $\lambda_0$ is large, then $\delta_n$ is small and vice versa.
>
> Q6: The robustness to outliers is somewhat evaluated by the adversarial attacks and also the newly included Gaussian noise injection, where square loss outperforms cross-entropy in almost all cases. Assumption C.3 is mainly technical and we do not think it relates much to robustness.
>
> ## Minor:
>
> Thank you for your comments and we have addressed them accordingly. You are correct that the expectation is taken with respect to $(X,Y)\sim P$. We have rewritten the corresponding terms into the form like$\mathbb{E}_{X\sim P_X}(1-\eta(X))\mathbb{I}\{f(X)\geq 0\}$ to make them more precise. We also changed the notation and plotted a new color bar.

---

> ### Author Response · Authors · 2021-11-15
> **Response to Reviewer 2eZX (Part 1)**
>
> Thank you for your comments.
>
> ## Regularization:
>
> We would like to point out that in the non-separable case, where labels contain noise, regularization is necessary. This is theoretically justified in the regression settings [Hu et al. 2021], and [Hu et al. 2019] applied regularization in the classification problem where the labels are randomly flipped. Most of the previous literature considers the separable case. When the classes are separable, one can set $\mu= 0$, which is also included in our Theorems 3.2 and 3.3, where we state$ \mu= o(1)$. In the synthetic data case, when two classes are non-separable, we observe that regularization could help with the accuracy improvement.
>
> ## NTK setting:
>
> We agree that the NTK setting may not be practical. Nevertheless, the presented theoretical guarantee does provide reassurance and new insights for the practical use of square loss. Specifically, the misclassification generalization bounds ensure consistent classification performance. The $L_\infty$ convergence rate on conditional probability estimation provides model calibration guarantee. The margin lower bound indicates robustness against (adversarial) perturbations. Our experiments also support our claims.
>
> ## Assumptions:
>
> We have added the discussions for the assumptions in Appendix D. It can be seen that most of them are standard in both the machine learning community and the statistics community.
>
> 1. Assumption D.1 is related to the neural network and GD training, where similar settings have been adopted by [Hu et al. 2021] and [Arora et al. 2019]. The requirements of the regularization parameter, the learning rate, the variance of initialization, the number of nodes in the hidden layer, and the iteration number are all the same as those in [Hu et al. 2021]. These assumptions are quite standard and used to show the consistency of the estimator under regression settings.
>
> 2. Assumption D.2 imposes conditions on the underlying true conditional probability in the *non-separable* case. This assumption basically requires that the conditional probability is representable by the GD-trained neural networks we consider (thus can be calibrated). Given that the neural networks are highly flexible, we believe that this is a mild condition. As a simple example, any Lipschitz functions satisfy this assumption.
>
> 3. Assumption D.3 requires that the complexity of the GD-trained neural network estimator is well-controlled. Since the step size is relatively small and the iteration number is not large (only $\log({\rm poly}(n,\xi, ,\lambda_0^{-1})$), we believe it is also a mild assumption.
>
> 4. Assumption D.4 only requires the probability to be upper bounded from infinity, while Assumption D.5 requires the probability to be upper bounded from infinity and lower bounded away from zero on the support $\Omega$. They are standard assumptions used in the classical analysis of classification in statistics; see [Audibert & Tsybakov 2007] and [Kohler & Krzyzak 2007] for example. Clearly, uniform distribution satisfies Assumptions D.4 and D.5.

---

### Official Review · Reviewer_Yckn · 2021-11-04

**Correctness:** 2
**Technical Novelty And Significance:** 1
**Empirical Novelty And Significance:** Not applicable
**Recommendation:** 3
**Confidence:** 3

**Main Review:**

My main concern about this submission is its relevance and importance. Most realistic experiments in this paper is on CIFAR-10/100, however I am not sure if the experiments included in the submission are convincing. Let me first describe my concern about the robustness experiments.

Authors measure the robustness of the square loss and cross entropy models using two kinds of adversarial attacks, PGD and AutoAttack. In general results using white-box adversarial attacks can be hard to interpret, as the performance of white-box attacks depend heavily on implementation details and attack hyperparameters. This was one reason historically that adversarial defense papers would get published in leading conferences only to be broken in less than a month. For the purposes of this submission, adversarial attacks are further concerning since the hyperparameters of these attacks were designed and optimized for cross-entropy loss, but the authors apply them to square loss here without re-optimizing them. I acknowledge that AutoAttack contains black-box attacks, such as SquareAttack, which may not suffer as much from this problem. However, it is not clear to me how much of the improvement of square loss on AutoAttack robustness is due to its improvement of SquareAttack. Furthermore, it is not clear to me if an improvement from 1% to 3% accuracy is significant, both of them seem low enough to be effectively useless. For robustness evaluation, I would recommend a simpler black-box noise such as Gaussian noise on the input with several noise scales. Another interesting experiment would be to evaluate CIFAR-10-C performance, which contains lots of noise-based corruptions that relate to the definition of robustness used in this submission.

Next concern is about the calibration results. It is nice to see that on the models tried by the authors, SL leads to an improved ECE. To evaluate the significance of this improvement, I would suggest including comparisons to ECE measures reported by published papers for the same model. This way the reader can see if the SL can lead to an improvement on ECE relative to the implementation of the same neural networks implemented by others. I think it would also be more helpful to train a more conventional and performant model such as WRN-28-10, for which there are plenty of available comparisons in literature.

**Summary Of The Paper:**

This paper studies square loss theoretically in the NTK regime. In this particular regime, authors find indications that square loss might lead to (adversarially) robust models and good calibration. Authors then run experiments outside of the NTK regime, on CIFAR-10 and CIFAR-100, to compare the robustness and calibration performance of square loss with cross-entropy.

**Summary Of The Review:**

Theoretical analysis is limited to toy models. Empirical studies should be improved demonstrate that square loss leads to more robust and calibrated models.

---

> ### Author Response · Authors · 2021-11-15
> **Response to Reviewer Yckn**
>
> ## General response
> This work proposes no new method for training classifiers and it is never our aim to improve the state-of-the-art performance in accuracy, robustness, and model calibration. We hope to provide more theoretical understanding of the square loss, by evaluating its **minimax convergence rate** and **boundary properties**. For DNN classifiers, one wouldn't expect them to only work well for complicated image data but not work under the much simpler classical setting. We hope theoretical results under the classical settings can provide reassurance of the practical use of square loss, thus we think they are relevant and important.
>
> We agree that the theoretical analysis is restricted in the case of one-hidden-layer neural networks, where we follow the standard settings of previous works in the NTK regime [Arora et al. 2019, Cao & Gu 2020, Nitanda & Suzuki 2020, etc.]. Nevertheless, the presented theoretical guarantee does provide new insights for the practical use of square loss. Specifically, the misclassification generalization bounds ensure consistent classification performance. The $L_\infty$ convergence rate on conditional probability estimation provides model calibration guarantee. The margin lower bound indicates robustness against (adversarial) perturbations.
>
> ## Robustness evaluation
> It is never our claim that square loss by itself is robust. What we argue is that it tends to be **more robust** than cross-entropy under the same settings. Hence, most of the comparisons are on the standard, baseline performances, without fancy bells and whistles. Indeed, from 1% to 5% may not be that significant but the same margin can be carried over to actual adversarial training. By substituting cross-entropy loss to square loss, the robust accuracy increased around **3%** while maintaining higher clean accuracy. See the table in the Overall Responses Part 3 for details.
>
> We appreciate the suggestion on Gaussian noise evaluation and we have tested our models accordingly. Gaussian noise with standard deviation from 0 to 0.4. As can be seen from the table below, SL usually outperforms CE, especially with Wide ResNet.
>
> |CIFAR-10||||||
> |-|-|-|-|-|-|
> ||0.00|0.10|0.20|0.30|0.40|
> |SL-ResNet18|95.04|**90.07**|**70.16**|**42.13**|**25.38**|
> |CE-ResNet18|**95.15**|90.03|69.71|41.08|24.66|
> |SL-WRN16-10|**95.02**|**88.49**|**60.91**|**35.78**|**24.04**|
> |CE-WRN16-10|93.94|84.78|56.63|33.70|22.41|
>
>
> |CIFAR-100||||||
> |-|-|-|-|-|-|
> ||0.00|0.10|0.20|0.30|0.40|
> |SL-ResNet50|**78.91**|**63.06**|**36.64**|**17.78**|**9.47**|
> |CE-ResNet50|79.82|62.72|34.42|16.69|9.11|
> |SL-WRN16-10|**79.65**|**62.01**|**30.69**|**15.11**|**8.88**|
> |CE-WRN16-10|77.89|60.14|26.47|10.26|5.57|
>
> ## Model calibration evaluation
>
> Thank you for the suggestion. To reiterate, this work proposes no new method for training classifiers and it is never our aim to improve the state-of-the-art performance in model calibration. What we argue is the superiority of square loss over cross-entropy in this regard. In this work, we only consider the baseline **vanilla** training and it's unfair to compare our results to existing literature specifically focusing on improving model calibration, where **extra fine-tune** procedures are usually involved.
>
> For the SOTA method on model calibration, we do think it's a good idea to evaluate whether the performance will improve if we substitute the base loss function to square loss. Similar experiments can be conducted for SOTA adversarial robust models but it's outside the scope of this work and deserves its separate empirical work.
>
>
> To summarize, we contribute to the theoretical understanding of using square loss to train deep classifiers and the motivation of our experiments is not to improve SOTA benchmarks, but to show square loss can be potentially better than cross-entropy in practice in certain aspects. Using square loss as the base classification loss function instead of cross-entropy, with modification and adaptions, may have higher potentials in model robustness and calibration.

---

### Author Response · Authors · 2021-11-15
**Overall Response (Part 3: Adversarial robustness)**

Stemming from the fact that $ 2\eta-1$ is much well-behaved than $\log \frac{\eta}{1-\eta}$ when classes are becoming more separable, we derived Theorem 3.3 stating the $l_2$ margin can be lower bounded. Since $l_\infty$ norm is lower bounded by $l_2$ norm (up to a constant multiplier), Theorem 3.3 naturally provides $l_\infty$ robustness towards adversarial attacks.

We utilize popular adversarial attacks to evaluate the classifier robustness. It is never our claim that square loss by itself is robust. What we argue is that it tends to be more robust than cross-entropy under the same settings. Indeed, from 1% to 5% may not be that significant but the same margin can be carried over to actual adversarial training. We have conducted several extra experiments using adversarial training. The table below lists results from standard PGD adversarial training with CE and SL. By substituting cross-entropy loss to square loss, the robust accuracy increased around 3% while maintaining higher clean accuracy.

||||||||
|-|-|-|-|-|-|-|
|CIFAR10|loss|acc %|attack|pgd steps|strength ($l_\infty$)|AutoAttack|
|Resnet18|CE|86.87|CE|3|8/255|37.08|
|||84.50||7|8/255|41.88|
|ResNet18|SL|**87.31**|angle|3|8/255|**40.46**|
|||**84.52**||7|8/255|**44.76**|

We have updated the submission PDF file. The main changes are:

- Cleaned up the notation and added discussion about the assumptions in Appendix D. Added an overview of the reproducing kernel Hilbert space.

- After the generalization error convergence rate results (Theorem 3.1 and 3.6), we added discussion and comparison to existing generalization error bounds.

- Added more discussion about the difference between square loss and cross-entropy, especially regarding their optimal solutions and ability to recover the conditional probability.

- Added more references. Specifically, we rewrote the simplex coding part (Section 4) and deleted claims that have been covered by existing literature, and made direct citations. We also added more references related to the neural network estimator within and out of the NTK regime (Appendix A).

- Added black-box Gaussian noise injection experiment to evaluate the classification robustness (Table G.3) as well as results for PGD adversarial training with square loss (Table 2).

---

### Author Response · Authors · 2021-11-15
**Overall Response (Part 2: Related work)**

We thank all the reviewers for the suggested references. There are numerous papers on classification and it's impossible to cover all of them. Nonetheless, we do want to discuss some **most relevant** literature to our work, about convergence rate and generalization error bound.

### Nonparametric Classification

[Mammen & Tsybakov 1999] first established optimal convergence rate for classifiers with smoothness assumption and Tsybakov's noise condition. **Sieve estimators** from minimizing the **0-1 loss** can achieve the optimal rate.

[Audibert & Tsybakov 2007] later considered plug-in estimators by minimizing the **square loss** and showed that **local polynomial estimators** can achieve the optimal convergence rate.

[Kohler & Krzyzak 2007] later extended the convergence results to **Nadaraya–Watson kernel estimators** (not related to kernel ridge regression). [Steinwart et al. 2007] studied **hinge loss** for **Gaussian kernel estimators** but got a sub-optimal convergence rate.

The aforementioned works are all on the non-separable case. Along this line, our work is the first to study how ONN classifiers work under square loss. Our analysis of ONN in the NTK regime has direct connections with **kernel ridge** ***regression*** (KRR). However, there are no existing *classification* convergence results for kernel-ridge-like classifiers under square loss. Even though our convergence rate is not optimal, it does provide **worst-case** guarantees justifying square loss paired with ONN as a theoretically valid classifier.

### Comparison between our generalization error bound (convergence rate) to existing ones:

Most of the existing generalization error bounds for deep classifiers are under the separable case and in the order of $O(1/\sqrt{n})$, which mainly depends on the complexity of the neural network, without much information about the ground truth. In comparison, our convergence rates are more general (for both separable case and non-separable case), and much faster. In the more extreme case of separable with a positive margin, the generalization error bound in [Ji & Telgarsky 2020] is also $O(1/\sqrt{n})$ but our bounds in Theorem 3.2 and 3.6 are exponentially fast.

### Reference:

[Mammen & Tsybakov 1999] Mammen, Enno, and Alexandre B. Tsybakov. "Smooth discrimination analysis." *The Annals of Statistics* 27.6 (1999): 1808-1829.

[Zhang 2004] Zhang, Tong. "Statistical behavior and consistency of classification methods based on convex risk minimization." *The Annals of Statistics* 32.1 (2004): 56-85.

[Bartlett et al. 2006] Bartlett, Peter L., Michael I. Jordan, and Jon D. McAuliffe. "Convexity, classification, and risk bounds." *Journal of the American Statistical Association* 101.473 (2006): 138-156.

[Audibert & Tsybakov 2007] Audibert, Jean-Yves, and Alexandre B. Tsybakov. "Fast learning rates for plug-in classifiers." *The Annals of Statistics* 35.2 (2007): 608-633.

[Steinwart et al. 2007] Ingo Steinwart, Clint Scovel, et al. Fast rates for support vector machines using Gaussian kernels. The Annals of Statistics, 35(2):575–607, 2007.

[Kohler & Krzyzak 2007] Michael Kohler and Adam Krzyzak. On the rate of convergence of local averaging plug-in classification rules under a margin condition. IEEE Transactions on Information Theory, 53(5):1735–1742, 2007.

[Kim et al. 2021] Kim, Yongdai, Ilsang Ohn, and Dongha Kim. "Fast convergence rates of deep neural networks for classification." *Neural Networks* 138 (2021): 179-197.

[Hu et al, 2019] Hu, Wei, Zhiyuan Li, and Dingli Yu. "Simple and effective regularization methods for training on noisily labeled data with generalization guarantee." *arXiv preprint arXiv:1905.11368*(2019).

[Ji & Telgarsky 2020] Ji, Ziwei, and Matus Telgarsky. "Polylogarithmic width suffices for gradient descent to achieve arbitrarily small test error with shallow relu networks." *arXiv preprint arXiv:1909.12292*(2019).

[Cao & Gu 2020] Cao, Yuan, and Quanquan Gu. "Generalization error bounds of gradient descent for learning over-parameterized deep relu networks." *Proceedings of the AAAI Conference on Artificial Intelligence*. Vol. 34. No. 04. 2020.

---

### Author Response · Authors · 2021-11-15
**Overall Response (Part 1: Cross-entropy vs Square loss)**

## Summary
This work contributes to the **theoretical** understanding of square loss in training overparametrized neural network (ONN) classifiers. In the classical smooth conditional probability classification setting, we derive the **first-ever convergence rate** for ONN classifiers using square loss trained by gradient descent (GD) and L2 regularization. For practical impact, we promote the use of square loss and find it to be advantageous in robustness and model calibration compared to cross-entropy.

## Cross-entropy vs Square loss
In statistics, binary classification with smoothness assumption has been extensively studied. Various surrogate losses to 0-1 loss have been investigated as well as the corresponding optimal classifiers. Let the ground-truth conditional probability be $\eta(x)$ and $\phi(f(x),y)$ be the surrogate loss, [Zhang 2004] listed the following well-established correspondence.

**Logistic loss:** $\phi(f(x),y)=\log(1+\exp(-yf(x)))$ $\longrightarrow$ **optimal** $f^* = \log\frac{\eta}{1-\eta}$

Hinge loss: $\phi(f(x),y)=\max(1-yf(x), 0)$ $\longrightarrow$ optimal $f^* = $$\mathrm{sign}(2\eta-1)$

**Square loss:** $\phi(f(x),y)=(1-yf(x))^2$ $\longrightarrow$ **optimal** $f^* = 2\eta-1$

CE estimates $\log(\eta/(1-\eta))$, which can be problematic when $\eta$ is close to one or zero (not even well-defined when $\eta=0$ or $\eta=1$). In comparison, square loss estimates $\eta$ directly, which is more well-behaved. From the optimal solution view, square loss has its uniqueness and can be advantageous in various perspectives, e.g., explicit feature modeling, model calibration, etc.

Take calibration error as an example, if $||f_W,a||_\infty < C$, which is often desired because we would like to control the complexity of the NN estimator. Hence, it cannot accurately estimate the conditional probability $\eta(x)$ where $\eta(x)>e^C/(1+e^C)$ or $\eta(x) < 1/(1+e^C)$. However, square loss does not have such a problem.

Theoretically, we have derived the 0-1 loss excess risk convergence rate for ONN classifiers under square loss. A similar convergence rate has **never** been developed under any other surrogate loss, due to tremendous **technical difficulty**. The reason why we can do it is the recently established correspondence between ONN trained with square loss using GD+weight decay and kernel ridge regression [Hu et al. 2021]. Given the correspondence, the analysis is still highly non-trivial. As for theoretical comparisons with CE, as much as we would like to develop similar convergence and boundary results for CE, we couldn't overcome the technical difficulties:

1) A sharp characterization of CE-trained neural networks under the non-separable case is still lacking.

2) As stated before, the neural network weights need to diverge in order to obtain a small calibration error, which introduces extra difficulties in the analysis of generalization and robustness.

Therefore, we turn to empirical evaluation for comparison. Through our numerical experiments, together with plenty of other empirical works, we found the square loss to have comparable generalization error but noticeable advantages in robustness and model calibration.

---

### Decision · Program_Chairs · 2022-01-20

**Decision:**

Reject

**Comment:**

The paper contributes a theoretical understanding of training over-parametrized deep neural networks using gradient descent with respect to square loss in the NTK regime. Besides giving guarantees on the classification accuracy using square loss, authors reveal several interesting properties in this regime including robustness and calibration.

The problem studied here is exciting and very relevant. The current version, unfortunately, has some shortcomings. For example, under a margin assumption, the authors show that the least-squares solution finds something with the margin and, therefore, it yields “robustness.” There is no quantification of how “robust” is the trained model, what is the threat model, what if the noise budget is larger than the attained margin. In general, the analysis lacks any careful finer characterization or quantification of the claimed properties. Besides, as was pointed out, the setting of the neural tangent kernel regime is somewhat limited and to some extent impractical. The assumptions under which the results hold further make the setting of the paper significantly restrictive.

The writing can be improved with more emphasis on the novelty and significance of the contributions. Currently, all of the assumptions are buried in the appendix and the main paper is not even self-contained. I believe the comments from the reviewers have already helped improve the quality of the paper. I encourage the authors to further incorporate the feedback and work towards a stronger submission.